# Changes in the Trunk and Lower Extremity Kinematics Due to Fatigue Can Predispose to Chronic Injuries in Cycling

**DOI:** 10.3390/ijerph18073719

**Published:** 2021-04-02

**Authors:** Alberto Galindo-Martínez, Alejandro López-Valenciano, Carlos Albaladejo-García, Juan M. Vallés-González, Jose L. L. Elvira

**Affiliations:** 1Sports Research Centre, Department of Sport Sciences, Miguel Hernández University, 03202 Elche, Spain; alberto.galindom@umh.es (A.G.-M.); calbaladejo@umh.es (C.A.-G.); jmi_vg@hotmail.com (J.M.V.-G.); 2Centre for Sport Studies, King Juan Carlos University, 28933 Madrid, Spain; alejandro.valenciano@urjc.es

**Keywords:** bike fitting, 3D kinematics, functional threshold power, statistical parametric mapping

## Abstract

Kinematic analysis of the cycling position is a determining factor in injury prevention and optimal performance. Fatigue caused by high volume training can alter the kinematics of the lower body and spinal structures, thus increasing the risk of chronic injury. However, very few studies have established relationships between fatigue and postural change, being these in 2D analysis or incremental intensity protocols. Therefore, this study aimed to perform a 3D kinematic analysis of pedaling technique in a stable power fatigue protocol 23 amateur cyclists (28.3 ± 8.4 years) participated in this study. For this purpose, 3D kinematics in hip, knee, ankle, and lumbar joints, and thorax and pelvis were collected at three separate times during the protocol. Kinematic differences at the beginning, middle, and end of the protocol were analyzed for all joints using one-dimensional statistical parametric mapping. Significant differences (*p* < 0.05) were found in all the joints studied, but not all of them occur in the same planes or the same phase of the cycle. Some of the changes produced, such as greater lumbar and thoracic flexion, greater thoracic and pelvic tilt, or greater hip adduction, could lead to chronic knee and lumbar injuries. Therefore, bike fitting protocols should be carried out in fatigue situations to detect risk factor situations.

## 1. Introduction

The kinematic analysis of a cyclist’s position during pedaling has become one of the most studied fields in cycling [1]. The influence of the position adopted by the cyclist on sports performance has been investigated, either through the power produced [2,3] or through aerodynamic position [4,5]. In addition, the lower body and trunk kinematics during pedaling have been analyzed [6,7] to detect different risk factors and reduce the most prevalent injuries in cycling [8].

The high training volume and intensity inherent to competition in cycling can generate high levels of fatigue in cyclists, triggering modifications in the kinematics of the lower body and trunk, which are structures responsible for bicycle propulsion [9]. Kinematic changes cause alterations in the magnitude and direction of the forces applied on the pedal, and consequently, a loss of performance [10,11] and a higher probability of cyclist injury [9]. Especially relevant are the kinematic variations caused by fatigue in the rachis structures [12] since they are the main torso stabilizers during pedaling [13]. It is suggested that keeping the rachis in a flexed position for long periods of time seems to be an influential factor in the occurrence of such a prevalent injury in cycling as low back pain (LBP) [14].

Moreover, a common strategy used to study pedaling kinematics has been to collect data within the 2D sagittal plane [1]. One primary problem with these analyses is that the frontal plane is mostly omitted [15], even though a combination of 3D kinematics including the frontal and transverse planes would provide information about the context surrounding the forces applied on the pedals. According to Pouliquen et al. [16], in order to maintain symmetric pedaling within the sagittal plane in fatigued situations, compensatory movements occur in other planes, which can lead to an increased risk of knee overuse injuries [17]. Therefore, it becomes essential to monitor these planes to understand kinematics changes more accurately due to fatigue and the associated risk of injury. Some of those movements, such as a greater knee displacement (projection) to the medial axis or a greater ankle flexion, which would generate a change in knee motion, may be related to knee injuries [18,19]. Knee injuries have received special attention from the literature in an attempt to prevent them due to the higher number of injuries in this joint [20].

In this regard, to the best of our knowledge, only three studies have focused on 3D motion during a fatigue protocol [11,12,16]. Pouliquen et al. [16] studied lower body kinematics in 12 elite cyclists in an incremental test. Moreover, Sayers et al. [11] and Sayers and Tweddle [12] recorded lower body and trunk kinematics in 10 amateur cyclists for one hour at a constant intensity (88% onset of blood lactate accumulation). Three studies showed significant changes in the main joints involved in the propulsion in cycling (hip, knee, and ankle). Another limitation of the literature, along with the paucity of studies, is that the studies that have analyzed the effect of fatigue on cycling kinematics have focused on the use of incremental tests to exhaustion [10,16,21]. In this regard, the changes in pedaling technique may be a consequence of an adaptation to applying greater power output [22]. In consequence, making use of maximal fatigue protocols of maintained intensity, such as the functional threshold power (FTP) test [23], seems to be a good alternative since the possible kinematic changes will come exclusively from accumulated fatigue and not from the combination of fatigue and intensity as in incremental tests [21]. Finally, most of the kinematic analyses (bike fitting) that are currently carried out on cyclists are performed in short protocols that do not induce fatigue, or they are performed on only one side of the body, despite the fact that bilateral asymmetries are common in cycling [24].

In addition, the studies focused on analyzing cycling kinematics are usually based on the paradigm of extracting information only from some discrete points of the pedaling cycle, commonly the lower and anterior position of the crank [25,26]. However, pedaling is a cyclic and continuous action, and therefore, this approach is clearly limited when comparing pedaling conditions. Analyses such as statistical parametric mapping (SPM) can detect differences in whole curves rather than just discrete points [27], allowing for a better understanding of the differences between conditions.

Hence, the objective of this study was to perform a continuous kinematic analysis of the pedaling technique in amateur cyclists during a maximal test of maintained intensity in order to observe the kinematic variations in the spinal and lower body joints due to fatigue.

## 2. Materials and Methods

### 2.1. Participants

The sample consisted of 23 amateur cyclists, 16 men (27.4 ± 9.6 years old, 1.77 ± 0.08 m, 72.0 ± 8.2 kg) and 7 women (33.3 ± 8.0 years old, 1.63 ± 0.05 m, 59.7 ± 8.6 kg), with at least three years of cycling experience (8.1 ± 4.7 years) and at least an average practice of 6 h/week (587 ± 220 min). In order to be included in the study, cyclists had to present no pain or pathologies that could modify the movement pattern in the six months prior to the study. All participants signed informed consent before their collaboration, based on the recommendations of the Helsinki declaration and approved by the institution’s Office of Responsible Research.

### 2.2. Procedure

A three-dimensional motion capture system consisting of seven T10 cameras and a Vero 2.2 (Vicon MX, Vicon Motion Systems Ltd., Oxford, UK), operating at 200 Hz, was used for kinematic analysis. To monitor pedaling conditions, the Wahoo KICKR Power Trainer potentiometer roller was used, with the Blue SC Wahoo cadence and speed sensor, validated by Zadow et al. [28], and the Wahoo^®^ Fitness: Workout Tracker APP on an Android system.

A lower limb and trunk model consisting of 45 markers was used. External reflective markers were placed on the L1, T6, and C7 vertebrae, both acromions, anterosuperior iliac spines, posterosuperior iliac spines, lateral and medial condyles of the femur, external and internal malleoli, calcaneus (lower and upper part), head of the first and fifth metatarsals, and toe. Additionally, technical markers were placed on the lateral end of the iliac crests and four markers clusters on the lateral side of each thigh and leg [29,30] (Figure 1). The markers on the anterosuperior iliac spines, internal femoral condyles, and internal malleoli acted as calibration markers to locate the hip, knee, and ankle joint centers, respectively.

Subsequently, a warm-up was performed by pedaling for 10 min at 100 W. This was followed by five steps of 1 min, with 25 W increments. In the last step, the aim was to reach an intensity close to that expected during the test. For this purpose, information was collected on previous FTP assessments of each cyclist. At the end of the warm-up, a 5 min rest was allowed. The warm-up was based on the recommendations of Bishop et al. [31], including low-intensity and high-intensity parts, followed by a rest before the start of the test.

After the rest, a maximal 20 min FTP test was performed for the assessment of the cyclists’ kinematics [23]. All participants used their own bicycles. Throughout the warm-up and the test, they were instructed to hold on to the middle handlebars so as not to vary the point of support and the kinematics of the spine and pelvis [32]. The pedaling cadence was monitored and maintained at around 90 rpm. To control for the maximum intensity of the test, the perception of effort (RPE) was recorded at the end of the test. The motion of the markers was recorded for 15 s in four specific moments, namely, after the start of the warm-up (WU), at the beginning of the test (INI), in the middle at 10 min (MID), and just before the end (FIN) [33]. From the 15 s recorded, the first five pedal strokes were discarded, and the next 10 complete pedal cycles of each lower limb were used for the study. Environmental conditions were controlled for all subjects, with constant temperature and humidity [34].

### 2.3. Data Processing and Analysis

Hip joint centers were calculated using the equation proposed by Harrington et al. [35] from the markers of the posterosuperior and anterosuperior iliac spines, the latter of which was reconstructed from the cluster formed by the markers of the posterosuperior spines and those of the iliac crests [36]. Moreover, the knee joint centers were located using the transepicondylar method [37,38]. The ankle joint center was found by calculating the intermediate point between the markers of the internal and external malleoli [39,40].

From the marker positions, hip, knee, and ankle joint angles were calculated three-dimensionally using conventional linear algebra procedures [41]. The angles of the pelvis and thorax segments were also calculated in 3D. Finally, the flexion angle of the lumbar region concerning the thorax was calculated in 2D. For the joint angles, a positive sign in x-, y-, and z-axis means flexion, abduction, and external rotation, respectively. For the segments, the positive sign means posterior tilt, right lateral rotation, and axial rotation to the left, respectively (Figure 2).

Additionally, we calculated the crank angle from the fifth metatarsal markers. The crank angle was used to slice the continuous data in every single pedaling cycle. Each cycle started when the ipsilateral pedal crossed the topmost position. The centered angles (lumbar, thorax, and pelvis) used the right pedal as a reference. All the angles were time normalized to 360 points with linear interpolation, each point representing 1° in a complete crank cycle.

### 2.4. Statistical Analysis

One dimensional SPM was used to compare time conditions along the pedaling cycle. To determine the effect of fatigue, INI, MID, and FIN periods were compared, performing an SPM repeated-measures ANOVA. In case a statistical difference was found, a post hoc pairwise t-test with Bonferroni correction was calculated. Cohen’s d [42] was also calculated to determine the magnitude of the paired differences. An alpha level of 0.05 was established for all analyses. To determine the effect of pedaling intensity, SPM paired t-test was calculated between WU and INI conditions.

All analyses were conducted using self-created scripts in Python 3.8 language, using the open source module spm1D for Python (v.0.4.3, www.spm1d.org (accessed on 20 December 2020)) [27].

## 3. Results

### 3.1. Effects of Fatigue

Figure 3 shows the results of the SPM repeated measures ANOVA comparing the three instants of the test (INI, MID, and FIN). Significant differences (*p* < 0.05) were shown in almost all the joints involved in pedaling. The differences in knee flexion–extension in the phase close to 0° of the crank, in ankle flexion–extension in the phase close to 0° and 180° of the crank, and in the adduction–abduction of both hips during the propulsion phase (50–100° of crank approximately) are noteworthy.

In the spinal area, differences were found in lumbar flexion–extension, during the initial propulsion phase (approximately 0–90° crank) of each lower extremity, and in thoracic flexion–extension throughout the pedal cycle. Concerning the lateral rotation, it increased in the thorax and pelvis segments, also corresponding to the beginning of the propulsion phase of each lower limb (0–90° of the crank). Finally, there was a rotation in the thorax, during the initial phase of the propulsion phase of the right lower limb (approximately 35° to 55° of the crank).

Table 1 also shows the specific pairwise comparisons when a statistical significance was found in the ANOVA. Some of the most important changes compared to the beginning of the test are the increase of ankle extension and knee flexion in the 0° crank phase, and changes in both hips in the y-axis in the propulsion phase, resulting in increased adduction. In the spinal structures, the most important differences are the greater flexion at the end of the test in the lumbar structure and a greater pelvic left lateral rotation when the left crank is at 180°.

### 3.2. Effects of Intensity

Finally, when comparing the two time points of the recording of different intensity (WU and INI), significant differences were revealed in both hips in the y-axis, with greater adduction at the beginning of the propulsion phase at the initial moment (0–130°) (left between 340° and 120° of the crank; *p* = 0.001; right between 10° and 10° of the crank; *p* = 0.001); and right between 10° and 140° of the crank; *p* < 0.001), and in the z-axis, with a greater external rotation at the initial moment, exclusive of the left hip in the final recovery phase (between 280–360° of the crank; *p* < 0.001) (Figure 4). We also found a greater thoracic flexion at the initial moment during the entire crank stroke (*p* = 0.001) and a greater homolateral pelvic tilt when the right crank was between 30° and 90° (*p* = 0.032) (Figure 4).

## 4. Discussion

This study aimed to analyze the kinematics of pedaling technique in amateur cyclists during a maximal test of maintained intensity. The main finding of this research was that there were significant changes in almost all the joints involved in pedaling throughout the maximal protocol. This confirms the limited validity of kinematic protocols, performed without fatigue, usually applied in the bike fitting context.

Differences in the lateral rotation of the thorax and pelvis segments (y-axis) may be the result of the cyclist pushing on the handlebars in the propulsion phase during pedaling [12]. On one hand, the greater trunk leans found at the final moment of the test may be associated with less coactivation of the trunk musculature, which increases the load on the spinal region and consequently the injury risk [43]. A greater lateral rotation, as has been shown in the thoracic region at the end of the fatigue protocol in this study, has also been previously associated with LBP risk in cycling [44].

On the other hand, the greater lumbar and thoracic flexion at the end of the protocol confirms the hypothesis that prolonged periods of holding a position produces a spinal creep [45], which could partly explain the high number of injuries suffered by professional cyclists in the spinal areas [46]. Srinivasan and Balasubramanian [47] have shown how increased fatigue in the trunk stabilizing muscles during pedaling, such as the erector spine, is often associated with cyclists presenting more severe symptoms in their LBP. Furthermore, people with LBP tend to have less trunk control [48]. The kinematic changes in the lumbar region occur at the end of the test and not in the middle, which confirms the need to develop fatigue protocols to detect the risk of LBP [49].

Regarding the lower body joints, the changes produced in the ankles are in line with other articles that studied the relationship between kinematics and fatigue [11,16,21]. We can assume that an increase in fatigue of the ankle extensor and flexor muscles causes an increase in joint stiffness [50] and consequently a change in the range of motion. The changes observed in the knee in the sagittal plane, in line with the results of Pouliquen et al. [16], are derived from the changes in the ankle angle, as they act in a closed kinematic chain [51]. However, this result differs from results found by other studies [11,21] in which the joint affected by the changes of the ankle’s sagittal plane was the hip. It has been suggested that the use of pedal cleats could modify the kinematics of the lower limb joints. Their use tends to produce changes in the knee and ankle, while changes occur in the knee and hip when cleats are not used [52]. Bini and Diefenthaeler [21] did use cleats, while Sayers et al. [11] did not specify. Among the cyclists, the use of cleats is widespread and we understand that most of the published studies are conducted with cleats. Another possible explanation is the participants’ choice of pedaling technique (pushing or pulling [53]), which is not usually reported and can have different joint implications. Finally, to our knowledge, this is the first study to apply a continuous statistical comparison of the pedaling technique not based just on discrete points, and therefore, they may not be entirely comparable.

Another key aspect is the change in the left knee, both in the y- and in the z-axis, with an increase in the adduction and the external rotation, and the increase in adduction in the hips in the y-axis. The kinematic alterations in the transverse and frontal planes could be related to the appearance of common cycling injuries such as patellofemoral pain [9,16,18]. Although this difference is only found in the left knee, it should not be assumed that the movement is cyclical and similar in both lower extremities [24]. As other studies have shown, it is common to find deficits in strength and range of motion between lower extremities [54]. This confirms the need to biomechanically analyze both legs during kinematic analyses, an aspect that is overlooked in some of the studies that have previously linked kinematics and injury [17,55].

Finally, changes in kinematics due to the intensity (WU vs. INI conditions) of pedaling in cyclists seem to justify the greater thoracic flexion, pelvis lateral rotation, and hip adduction at the beginning of the protocol compared to the warm-up, as an adaptation to performing greater power [22]. To the authors’ knowledge, this is the first study to analyze 3D kinematic differences in a constant intensity fatigue protocol. Furthermore, this study presents a larger sample size than previous studies conducted in 3D [11,12,16]. Therefore, the results obtained show the need to develop kinematic analysis protocols, adapted to the specific fatigue demands of each cycling discipline. This allows the optimization of movement to avoid injuries and improve performance and highlights the need to adopt preventive programs for the joints mainly affected by fatigue. Future research should investigate the relationship of core stability/strength with consistency in the spine and lower body kinematics. According to Abt et al. [56], core stability reduces torso movement and lower body alignment, and this could reduce the risk of injury in cyclists. Asplund and Ross [57] suggested that greater core stability would reduce the risk of suffering a spine injury, and this could be achieved through training focused on the trunk, performing dynamic and/or static exercises. One of the newest methods for quantifying stability in static exercises is accelerometry, which would allow us to monitor the status of core stability and improve the dose–response in programs focused on the trunk [58].

The main limitations of this study have to do with the heterogeneity of the sample since, even though they were amateur cyclists, they reported differences in the volume of weekly minutes of training and in their experience, a variable that could influence the kinematic changes. An experienced cyclist may have greater consistency in kinematics [59]. Another important limitation is the absence of an external recording of physiological variables associated with fatigue, such as heart rate or blood lactate, as the RPE may be insufficient in people unfamiliar with the scale.

## 5. Conclusions

A maximum fatigue protocol at stable intensity appears to modify the kinematic pattern of amateur cyclists’ spinal, pelvic, and lower body structures. This modification in kinematics may increase the risk of injury to these structures and reduce performance. For this reason, it is important to carry out bike-fitting protocols in fatigue situations, which are close to the specific demands of competition and allow the detection of possible risk factors for injury. Additional trunk stability training is likely to reduce the movement associated with fatigue, thereby reducing the risk of injury.

## Figures and Tables

**Figure 1 ijerph-18-03719-f001:**
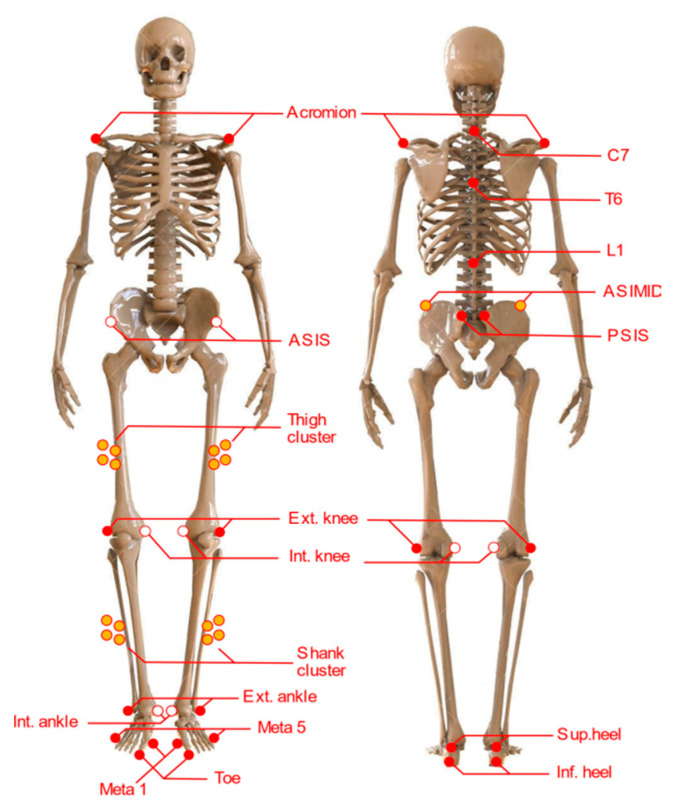
Anterior and posterior view of the location of the markers of the model used to capture the movement of the body. Filled with orange technical markers. Filled with white calibration markers.

**Figure 2 ijerph-18-03719-f002:**
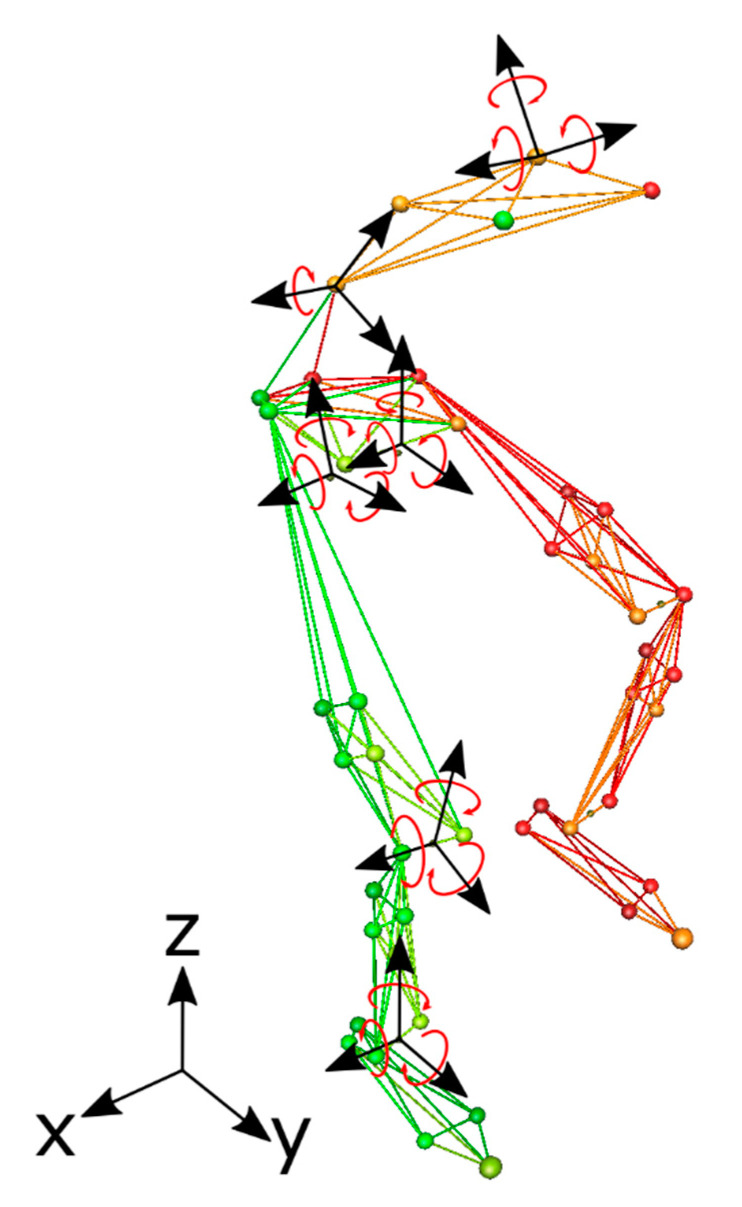
View of the local coordinate system of each calculated joint and segment angles. The curved arrows indicate the positive sign of each joint or segment axis.

**Figure 3 ijerph-18-03719-f003:**
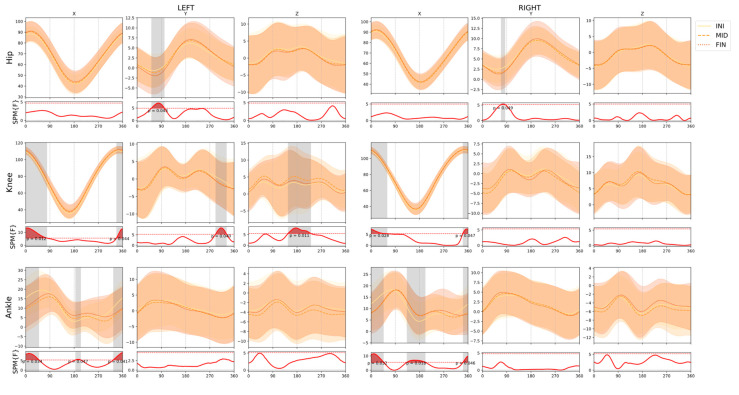
Comparison of the joint and segment angles across the time in the test. Lower panels display statistical parametric mapping F values from the repeated measures ANOVA, with the dotted line representing the critical threshold. Red and grey shaded areas indicate statistical differences among conditions. Horizontal values represent one pedaling cycle, starting from the top position of the same side pedal. INI: measurement at 0 min, MID: measurement at 10 min, FIN: measurement at 20 min. SPM: statistical parametric mapping. x, y, z: movement in the sagittal, frontal, and transverse plane, respectively.

**Figure 4 ijerph-18-03719-f004:**
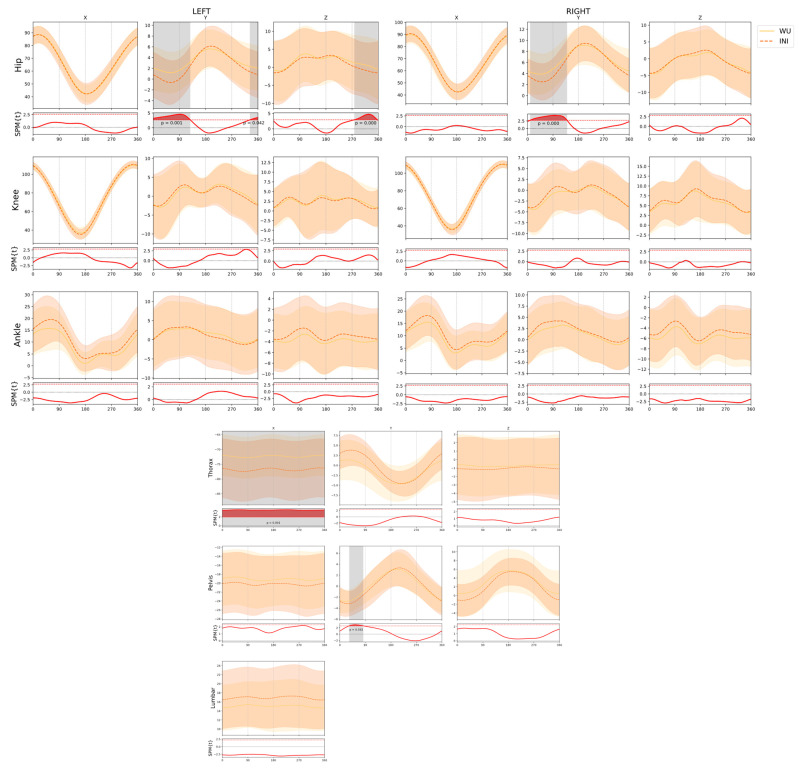
Comparison of the joint and segment angles between the warm-up and the initial part of the test. Lower panels display statistical parametric mapping t values from the t-test, with the dotted line representing the critical threshold. Red and grey shaded areas indicate statistical differences among conditions. Horizontal values represent one Pedaling cycle, starting from the top position of the same side pedal. WU: measurement when starting the warm-up. INI: measurement at 0 min of the test. SPM: statistical parametric mapping. x, y, z: movement in the sagittal, frontal, and transverse plane, respectively.

**Table 1 ijerph-18-03719-t001:** Post hoc comparisons between the ANOVA conditions of the functional threshold power (FTP) representing the joints axis and the clusters where these differences were found. INI: measurement at 0 min, MID: measurement at 10 min, FIN: measurement at 20 min. x, y, z: movement in the sagittal, frontal, and transverse plane, respectively.

Condition	Joint and Axis	Cluster (Crank Angle)	*p*	Effect Size
INI vs. MID	Left Ankle_x	[329, 49]	0.044	0.51
	Right Ankle_x	[341, 46]	<0.01	0.44
		[133, 201]	<0.01	0.30
	Right Hip_y	[68, 83]	0.022	0.31
	Left Knee_x	[339, 39]	0.045	0.31
	Right Knee_x	[341, 59]	0.018	0.24
	Left Knee_y	[291, 331]	0.023	0.18
	Left Knee_z	[146, 231]	<0.01	0.19
	Pelvis_y	[0, 13]	0.049	0.40
	Thorax_y	[172, 267]	0.035	0.65
	Thorax_z	[32, 53]	0.016	0.22
INI vs. FIN	Left Ankle_x	[337, 31]	0.040	0.49
		[184, 204]	<0.01	0.52
	Left Hip_y_	[53, 101]	<0.01	0.31
	Right Hip_y	[68, 83]	0.017	0.36
	Left Knee_x	[338, 78]	0.045	0.28
	Right Knee_x	[0, 33]	0.033	0.24
	Left Knee_y	[291, 332]	0.011	0.20
	Lumbar_x	[44, 60]	0.049	0.39
		[202, 242]	0.047	0.38
	Pelvis_y	[336, 63]	<0.01	0.40
	Thorax_y	[8, 71]	<0.01	0.48
		[159, 277]	<0.01	0.70
	Thorax_z	[32, 53]	0.036	0.31
MED vs. FIN	Left Knee_z	[158, 230]	<0.01	0.14
	Thorax_x	[347, 116]	0.049	0.31
		[204, 262]	0.047	0.31
	Thorax_y	[8, 70]	0.033	0.39

## Data Availability

The data presented in this study are available on request from the corresponding author.

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
