# Peer review of "Changes in the Trunk and Lower Extremity Kinematics Due to Fatigue Can Predispose to Chronic Injuries in Cycling"

_ijerph, 2021, doi:10.3390/ijerph18073719_

Round 1

Reviewer 1 Report

General Comments

The paper addresses a useful question on the effect of fatigue on lower body and spine 3D kinematics during a maximal fatigue cycling protocol at a stable power intensity. The study falls within the scope of suggesting evaluation protocols for bike fitting that can be closer to the specific demands of the cycling activity as well as more effective in injury prevention. The information provided in the Introduction section was clear and justified the methodological as well as the innovative aspects of the study. The authors adopted the Statistical Parametric Mapping method to investigate changes in the kinematics of joints and segments in all 3 spatial planes and along the entire pedalling cycle, which makes the methodology basis of the present work novel and very interesting. Results indicated that kinematic changes are mainly observed at the level of trunk segment, ankle and knee joints at specific phases of the pedalling cycle, with changes being already visible from the middle of the fatiguing protocol. The results are coherently presented and discussed within the framework of cycling performance and overuse injury prevention. Overall, the manuscript is well-written and only minor changes are needed in order to consider the paper for publication.

Minor Issues

Line 60: no need of OBLA acronym definition as it is not used in the rest of the manuscript

Line 62: “Another limitation…” not clear what was the first limitation of the previously reported literature

Line 105: please check the reference format

Line 112: “10 min at 100 watts” please justify the choice of the starting power output, was it based on a percentage of previous tests? Was there any gender-related difference?

Lines 208-210: please check the Figure 4 caption to improve clarity.

Lines 220-223: please rephrase the sentence for better clarity.

Lines 240-241: “However, this result differs from results found by other studies [11,21] where the joint affected by the changes of the ankle’s sagittal plane was the hip”, authors should attempt to explain the possible reason behind such difference as they did for the literature that is in agreement with the present results.

Author Response

Dear reviewer,

We submitted our reply in the attached word file.

Reviewer 2 Report

In the present article Alberto Galindo-Martínez et al. attempted to explore how a maximum fatigue protocol at stable intensity appears to modify the kinematic patern of amateur cyclists’ spinal, pelvic and lower body structures. Their findings suggest that this alteration in kinematics may increase the risk of injury to these structures. For this reason, it is important to carry out bike fitting protocols in fatigue situations, which are close to the specific demands of competition and allow the detection of possible riskfactors for injury.  The study is potentially interesting and the methodologies that have been used are appropriate and show a good knowledge of the correct experimental approach within the in vivo kinematics studies exploration techniques and data analyses. The discussion section can be ameliorated, better integrating the findings of the work with available literature. Nonetheless, in the current form, the result appears as a short, concise report that should be expanded in order to enhance the novelties, discuss and substantiate the main observations to support the final hypothesis explaining what protocols the autors would adopt to perform an additional trunk stability training in order to reduce the movement associated with fatigue, thereby reducing the risk of injury; however, it could also be an interesting starting point for a subsequent work.  Furthermore, the overall level of English should be improved. 

Author Response

(The authors gave the same response as above.)
